# Modeling and benchmarking quantum optical neurons for efficient neural computation

Andrea Andrisani[1]☯, Gennaro Vessio 💿[2]*☯, Fabrizio Sgobba[3], Francesco Di Lena[1], Luigi Amato Santamaria[1]☯, Giovanna Castellano 💿[2]☯

1 Giuseppe Colombo Centre for Space Geodesy, Italian Space Agency, Matera, Italy, 2 Department of Computer Science, University of Bari Aldo Moro, Bari, Italy, 3 Department of Physics, University of Bari Aldo Moro, Bari, Italy

☯ These authors contributed equally to this work.
* gennaro.vessio@uniba.it

## Abstract

Quantum optical neurons (QONs) are emerging as promising computational units that leverage photonic interference to perform neural operations in an energy-efficient and physically grounded manner. Building on recent theoretical proposals, we introduce a family of QON architectures based on Hong–Ou–Mandel (HOM) and Mach–Zehnder (MZ) interferometers, incorporating different photon modulation strategies—phase, amplitude, and intensity. These physical setups yield distinct pre-activation functions, which we implement as fully differentiable software modules. We evaluate these QONs both in isolation and as building blocks of multilayer networks, training them on binary and multiclass image classification tasks using the MNIST and FashionMNIST datasets. Each experiment is repeated over five independent runs and assessed under both ideal and non-ideal conditions to measure accuracy, convergence, and robustness. Across settings, MZ-based neurons exhibit consistently stable behavior—including under noise—while HOM amplitude modulation performs competitively in deeper architectures, in several cases approaching classical performance. In contrast, phase- and intensity-modulated HOM-based variants show reduced stability and greater sensitivity to perturbations. These results highlight the potential of QONs as efficient and scalable components for future quantum-inspired neural architectures and hybrid photonic–electronic systems. The code is publicly available at https://github.com/gvessio/quantum-optical-neurons.

## 1 Introduction

Artificial intelligence (AI) based on large-scale neural networks has achieved remarkable success in various fields, including computer vision, language processing, and pattern recognition. This progress is primarily due to the availability of massive datasets and increased computational power. However, the performance of such models

**Data availability statement:** All datasets used in this study are publicly available. The MNIST dataset is available at: http://yann.lecun.com/exdb/mnist/. The FashionMNIST dataset is available at: https://github.com/zalandoresearch/fashion-mnist. Both datasets are also directly accessible through the PyTorch ecosystem via the torchvision.datasets module.

**Funding:** This study was supported by the Italian Ministry of University and Research in the form of a grant awarded to GC (Project CLOSE – ARS01_00141). The specific roles of this author are articulated in the 'author contributions' section. The funders had no role in study design, data collection and analysis, decision to publish, or preparation of the manuscript.

**Competing interests:** The authors declare that they have no competing interests.

comes at the cost of significant energy consumption, prompting interest in alternative hardware architectures that are both efficient and scalable.

One promising direction is optical computing, where information is processed using light instead of electricity [1,2]. Optical systems can perform key neural operations, such as linear combinations and nonlinear activations, using components like interferometers, modulators, and lenses [3–6]. These systems can, in principle, implement artificial neurons with high speed and low power consumption [7].

In parallel, quantum machine learning has investigated how quantum systems can efficiently perform learning tasks by exploiting entanglement [8–11]. Approaches based on physical systems that emulate qubits, such as Bose–Einstein consensation [12,13], spin glasses [14] or ensembles of polarized photons [9,15], have been shown to potentially implement algorithms for segmentation or Boltzmann machine training with an exponential speed-up compared to their classical counterparts. More recently, quantum kernels evaluated through Hong–Ou–Mandel (HOM) interference [16–23] between pairs of time- or frequency-modulated photons—effectively encoding qu-words—have been proposed [24]. Remarkably, despite relying on a minimal system composed of only two particles, this approach still achieves exponential speed-up.

A further advance toward more efficient AI was presented in Roncallo et al. [25], where the concept of quantum optical neuron (QON) was introduced. In this framework, input and weight vectors are encoded into single-photon wavefronts using spatial light modulators. The scalar product between these quantum states is then estimated through HOM interference, enabling a hardware-level realization of a neuron's pre-activation—namely, the internal combination of inputs and weights prior to the activation function. Significantly, this approach achieves a computational cost independent of input dimensionality (super-exponential speed-up) while exploiting the inherently bosonic nature of photons.

In this work, we extend and generalize the QON model by exploring new optical configurations, particularly a Mach–Zehnder (MZ) [26–33] interferometer setup with dual modulation paths. We investigate amplitude-, phase-, and intensity-based modulation schemes, yielding a broader family of physically realizable pre-activation functions that differ nonlinearly from classical inner products.

Unlike quantum kernel methods [24,34], our approach does not involve high-dimensional Hilbert space embeddings. Instead, we focus on quantum-inspired functions that are experimentally implementable and potentially more efficient in training and inference. QONs are simulated here in software as fully differentiable modules, enabling end-to-end training with gradient-based optimization. We also introduce a layered quantum optical neural network (QONN) architecture built entirely from QONs, and demonstrate its ability to scale to multiclass classification problems.

Beyond expanding the family of physically grounded pre-activation functions, we provide a systematic and statistically robust assessment of QON performance. All experiments are repeated across multiple independent runs and evaluated under both ideal and non-ideal conditions, allowing us to quantify accuracy, convergence stability, and sensitivity to realistic optical imperfections. Performing simulations under

non-ideal conditions, accounting for decoherence, noise, and photon losses, could yield valuable insights for the practical implementation of a quantum artificial neuron based on photons. In this regard, we mention that quantum machine learning implemented with photon technologies has already been experimentally verified in [35].

Our contributions can be summarized as follows:

- We develop a unified theoretical and computational framework for QONs based on HOM and MZ interferometric architectures.

- We derive a family of physically grounded pre-activation functions and integrate them into modular, fully differentiable QON and QONN designs.

- We provide a systematic and statistically robust comparison of QON variants—under both ideal and non-ideal conditions and relative to a classical baseline—clarifying their accuracy, stability, and robustness properties.

These results support the idea that QONs are not only theoretically interesting but also practically viable for future low-power and high-speed AI applications, including autonomous vision, wearable health monitoring, remote sensing, and fast scientific imaging.

## 2 Mathematical framework for quantum optical neurons

### 2.1 Model and pre-activation

Quantum optical neurons, introduced by Roncallo et al. [25], represent a paradigm in which the input data and weight parameters are independently encoded onto the quantum states of two photons. These states interfere within a quantum optical system, and the resulting interference pattern defines the neuron's pre-activation. Remarkably, this mechanism enables a computational cost that does not scale with the number of parameters, as the scalar product is physically estimated through photon interference.

Let $\boldsymbol{\mu} = (\mu_1, \ldots, \mu_N) \in \mathbb{R}^N$ and $\boldsymbol{\lambda} = (\lambda_1, \ldots, \lambda_N) \in \mathbb{R}^N$ denote input and weight vectors, being $N$ the input dimensionality. Light modulators apply transformations $\hat{O}_{\boldsymbol{\mu}}$ and $\hat{O}_{\boldsymbol{\lambda}}$ on an initial photon state $|\eta\rangle$, yielding two states:

$$|\psi\rangle = \hat{O}_{\boldsymbol{\mu}} |\eta\rangle = \sum_{j=1}^{N} v_j(\boldsymbol{\mu}) |\alpha_j\rangle,$$

$$|\phi\rangle = \hat{O}_{\boldsymbol{\lambda}} |\eta\rangle = \sum_{j=1}^{N} w_j(\boldsymbol{\lambda}) |\alpha_j\rangle,$$

$$(1)$$

where $\{|\alpha_j\rangle\}_{j=1}^{N}$ is an orthonormal photon state basis. The scalar product of these states defines the pre-activation:

$$\langle\phi|\psi\rangle = \sum_{j=1}^{N} w_j^*(\boldsymbol{\lambda}) v_j(\boldsymbol{\mu}).$$

$$(2)$$

Depending on the optical setup, the output activation can take two forms:

$$f_1\left(\langle\phi|\psi\rangle\right) = h\left(\left|\langle\phi|\psi\rangle\right|^2 + b\right),$$

$$(3)$$

or alternatively,

$$f_2\left(\langle\phi|\psi\rangle\right) = h\left(a_R \operatorname{Re}\langle\phi|\psi\rangle + a_I \operatorname{Im}\langle\phi|\psi\rangle + b\right),$$

$$(4)$$

where $h$ denotes the activation function (e.g., sigmoid), $b$ is a bias term, and for (4), $a_R$ and $a_I$ are two additional scalar hyperparameters to be fixed.

For comparison, in conventional artificial neurons, the output activation is:

$$f(\boldsymbol{\lambda}, \boldsymbol{\mu}) = h\left(\boldsymbol{\lambda} \cdot \boldsymbol{\mu} + b\right),$$

(5)

where the scalar product $\boldsymbol{\lambda} \cdot \boldsymbol{\mu}$ has a cost that scales linearly with $N$.

Roncallo et al. [25] proposed a HOM interferometer to measure $|\langle\phi|\psi\rangle|^2$. In Section 3.1, we show that a Mach–Zehnder interferometer allows a broader range of pre-activations with potentially greater flexibility. Indeed, with MZ bunching probabilities, we can recover $\text{Re}\{\langle\phi|\psi\rangle\}$ and $\text{Im}\{\langle\phi|\psi\rangle\}$ separately, in place of $|\langle\phi|\psi\rangle|^2$ as in the HOM interference case, thus allowing output expression estimation like (4).

## 2.2 Gradient evaluation and backpropagation

Training a QON involves computing the gradient of $f$ with respect to $\boldsymbol{\lambda}$. For the output form (3), the gradient is:

$$\frac{\partial}{\partial\boldsymbol{\lambda}} f_1 = h'\left(|\langle\phi|\psi\rangle|^2 + b\right) \text{Re}\left\{\langle\phi|\psi\rangle \left\langle \frac{\partial\phi}{\partial\boldsymbol{\lambda}}|\psi\right\rangle^*\right\}.$$

(6)

For (4), we have:

$$\frac{\partial}{\partial\boldsymbol{\lambda}} f_2 = h'\left(a_R\text{Re}\,\langle\phi|\psi\rangle + a_I\text{Im}\,\langle\phi|\psi\rangle + b\right) \cdot$$
$$\cdot \left(a_R\text{Re}\left\langle\frac{\partial\phi}{\partial\boldsymbol{\lambda}}|\psi\right\rangle + a_I\text{Im}\left\langle\frac{\partial\phi}{\partial\boldsymbol{\lambda}}|\psi\right\rangle\right).$$

(7)

The main computational effort lies in evaluating the term $\left\langle\frac{\partial\phi}{\partial\boldsymbol{\lambda}}|\psi\right\rangle$, whose form depends on the chosen photon modulation strategy. This term cannot be obtained through straightforward physical measurements, such as $|\langle\phi|\psi\rangle|^2$ via a HOM interferometer, as demonstrated in [25], or $\langle\phi|\psi\rangle$ using a MZ interferometer, which we will discuss in Section 3.1. Instead, it must be computed mathematically.

In a classical artificial neuron, the gradient takes the well-known form:

$$\frac{\partial}{\partial\boldsymbol{\lambda}} f = h'\left(\sum_{j=1}^{N} \lambda_j\mu_j + b\right)\boldsymbol{\mu}.$$

(8)

The computational cost of (8) scales with $N$, as there are $N$ derivatives that need to be calculated. The computation begins with the first derivative, which has a cost of $\mathcal{O}(N)$. However, each of the subsequent derivatives can be computed at a cost that is independent of $N$. This is because $h'\left(\sum_{j=1}^{N} \lambda_j\mu_j + b\right)$ serves as a common factor among all the derivatives.

When QONs are embedded in deeper architectures, as discussed in Section 3.3, additional gradient terms appear due to the dependencies between layers—for instance, the derivatives with respect to both $\boldsymbol{\lambda}$ and $\boldsymbol{\mu}$ must be propagated through successive QON modules.

## 2.3 Photon state modulation

The effectiveness and feasibility of QONs depend critically on how $\boldsymbol{\lambda}$ and $\boldsymbol{\mu}$ are encoded onto photon states. This process is performed via light modulators, which apply specific amplitude and/or phase transformations to a reference photon wavefront.

In this section, we describe two primary strategies for modulation: amplitude modulation and phase modulation in the space domain. Each leads to a different mathematical form of the scalar product $\langle\phi|\psi\rangle$ and its gradient, ultimately influencing the dynamics of training.

Modulation in the spatial domain can be achieved using spatial light modulators (SLMs), which locally alter the amplitude and/or phase components of the photon wavefronts. These modifications occur uniformly within small regions of the wavefront—denoted as pixels—which define the vector basis $\{|\alpha_j\rangle\}_{j=1}^N$ associated with the SLM action in (1). If the incoming wavefronts are plane and the pixel dimensions are much larger than the photon wavelength, diffraction effects can be neglected. In this case, for $|\alpha_j\rangle$ one can write:

$$\langle x, y|\alpha_j\rangle \equiv \theta_{L,j}(x,y) = \theta_L\left(\left|x-x_j\right|, \left|y-y_j\right|\right),$$

(9)

where $\theta_L$ is the normalized two-dimensional rectangular box function centered at the origin with side length $L$, and $(x_j, y_j)$ are the pixel centers. We assume that photon wavefronts lie on a plane parallel to the $(x, y)$ plane, with pixels of rectangular shape. The state functions in (9) form an orthonormal set.

When the pixel dimensions become comparable to the photon wavelength, diffraction effects can no longer be ignored, and (9) must be interpreted as a boundary condition for the propagation of the electromagnetic wave functions in the interference region. However, if photon interference occurs in the far-field regime—either at sufficiently large distances from the SLMs or at the common focal plane of two imaging lenses with the SLMs as objects—the states $|\alpha_j\rangle$ are given by the Fourier transform of the right-hand side of (9), in accordance with Fraunhofer diffraction. Without loss of generality, (9) can still be used in the far-field limit, since by Parseval's theorem the scalar products between two functions remain unchanged when the functions are replaced by their Fourier transforms, up to a constant multiplicative factor.

**2.3.1 Amplitude/intensity modulation.** In amplitude modulation, the local amplitude of the photon's wavefront inside each pixel is scaled according to the corresponding entry of the input or weight vector. Taking into account the normalization condition of quantum states, a natural choice is to set each vector to unit $\ell_2$-norm:

$$\boldsymbol{v} = \frac{\boldsymbol{\mu}}{\|\boldsymbol{\mu}\|}, \qquad \boldsymbol{w} = \frac{\boldsymbol{\lambda}}{\|\boldsymbol{\lambda}\|},$$

(10)

so that the modulated quantum states yield the scalar product:

$$\langle\phi|\psi\rangle = \sum_j \frac{\lambda_j \mu_j}{\|\boldsymbol{\lambda}\|\|\boldsymbol{\mu}\|}.$$

(11)

The derivative of (11) with respect to $\lambda_k$ is:

$$\left\langle \frac{\partial\phi}{\partial\lambda_k}|\psi\right\rangle = \frac{\mu_k}{\|\boldsymbol{\mu}\|\|\boldsymbol{\lambda}\|} - \frac{\lambda_k}{\|\boldsymbol{\lambda}\|^2}\langle\phi|\psi\rangle.$$

(12)

Alternatively, if intensities (rather than amplitudes) are modulated, the quantum state components become proportional to the square roots of vector entries. In this case, the scalar product reads:

$$\langle\phi|\psi\rangle = \sum_j \sqrt{\frac{\lambda_j \mu_j}{|\boldsymbol{\lambda}|_1 |\boldsymbol{\mu}|_1}}.$$

(13)

The corresponding gradient is:

$$\left\langle \frac{\partial \phi}{\partial \lambda_k} | \psi \right\rangle = \frac{1}{2} \sqrt{\frac{\mu_k/\lambda_k}{|\boldsymbol{\mu}|_1 |\boldsymbol{\lambda}|_1}} - \frac{\langle \phi | \psi \rangle}{2|\boldsymbol{\lambda}|_1}.$$

(14)

Observe that since $\lambda_j$ and $\mu_j$ are positive values—associated with the pixel transmission coefficients in the SLMs—the following relationship holds:

$$\langle \phi | \psi \rangle = |\langle \phi | \psi \rangle| = \sqrt{|\langle \phi | \psi \rangle|^2}.$$

(15)

This means that $\langle \phi | \psi \rangle$ can be obtained from $|\langle \phi | \psi \rangle|^2$. Consequently, MZ interference measurements can also be achieved using HOM interference. Therefore, when limited to amplitude or intensity modulations, HOM interference is preferable to MZ interference, which requires a more complex hardware setup.

**2.3.2 Phase modulation.** In phase modulation, each component of the photon's wavefront is assigned a phase based on the input or weight vector. A typical encoding uses:

$$v_j = \frac{e^{i\mu_j}}{\sqrt{N}}, \qquad w_j = \frac{e^{i\lambda_j}}{\sqrt{N}},$$

(16)

so that the scalar product becomes:

$$\langle \phi | \psi \rangle = \frac{1}{N} \sum_{j=1}^{N} e^{i(\mu_j - \lambda_j)}.$$

(17)

About the linear combination $a_R \text{Re} \langle \phi | \psi \rangle + a_I \text{Im} \langle \phi | \psi \rangle$ appearing in (4), it reads:

$$a_R \text{Re} \langle \phi | \psi \rangle + a_I \text{Im} \langle \phi | \psi \rangle = \frac{A}{N} \sum_{j=1}^{N} \cos(\mu_j - \lambda_j + \theta),$$

(18)

where $A = \sqrt{a_R^2 + a_I^2}$ and $\theta = \arctan(a_I/a_R)$. In the following, we set $A = 1$ and

$$a_R = \cos \theta, \qquad a_I = \sin \theta,$$

(19)

and just leave a tunable phase offset $\theta$.

The gradients in this case are:

$$\left\langle \frac{\partial \phi}{\partial \lambda_k} | \psi \right\rangle^* = i e^{i(\lambda_k - \mu_k)},$$

(20)

$$\text{Re} \left\langle \frac{\partial \phi}{\partial \lambda_k} | \psi \right\rangle \cos \theta + \text{Im} \left\langle \frac{\partial \phi}{\partial \lambda_k} | \psi \right\rangle \sin \theta = \frac{1}{N} \sin(\mu_k - \lambda_k + \theta).$$

(21)

## 3 Mach–Zehnder optical neurons and quantum optical neural networks

In the original QON model proposed by Roncallo et al. [25], the Hong–Ou–Mandel effect is used to define the pre-activation as the squared overlap between two modulated quantum states:

$$\mathcal{T}_1 = \left|\langle\phi|\psi\rangle\right|^2 + b, \tag{22}$$

which appears as an argument of $h$ in (3), where $|\phi\rangle$ and $|\psi\rangle$ are single-photon states entering the two input ports of a beam splitter. The probability of detecting two photons in separate output ports (coincidence event) is given by:

$$P_{\text{coinc.}} = \frac{1 - \left|\langle\phi|\psi\rangle\right|^2}{2}, \tag{23}$$

so that the pre-activation function $\mathcal{T}_1$ can be achieved by a quantum measurement with minimal computational cost:

$$\mathcal{T}_1 = 1 - 2P_{\text{coinc.}} + b. \tag{24}$$

On the other hand, with a HOM-based QON, it is not possible to efficiently reproduce the pre-activation function

$$\mathcal{T}_2 = \text{Re}\{\langle\phi|\psi\rangle\}\cos\theta + \text{Im}\{\langle\phi|\psi\rangle\}\sin\theta + b, \tag{25}$$

which appears by (19) as the argument of $h$ in (4), since $\text{Re}\{\langle\phi|\psi\rangle\}$ and $\text{Im}\{\langle\phi|\psi\rangle\}$ cannot be directly extracted from $P_{\text{coinc.}}$ and would instead require calculations with computational cost $\mathcal{O}(N)$.

In this section, we extend the concept by employing a Mach–Zehnder interferometer. This alternative configuration allows the full recovery of both the real and imaginary parts of the scalar product $\langle\phi|\psi\rangle$ directly from photon detection statistics, thereby enabling an efficient implementation of the pre-activation function $\mathcal{T}_2$. After comparing the computational costs of the two QON architectures, we finally illustrate how such neurons can be employed as building blocks for multi-layer quantum optical neural networks.

### 3.1 Mach–Zehnder-based quantum optical neuron

The core component of the MZ QON is the interferometer shown in Fig 1. A two-photon state $|\Psi\rangle$, where both photons are initially in the same state $|\xi\rangle$, enters port 1 of the first beam splitter:

$$|\Psi\rangle = \frac{1}{\sqrt{2}}a_1^\dagger a_1^\dagger |\text{vac}\rangle, \tag{26}$$

where $a_1^\dagger$ creates a photon in the state $|\eta\rangle$ at port 1.

Assuming a symmetric 50:50 beam splitter with transfer matrix

$$\frac{1}{\sqrt{2}}\begin{bmatrix} 1 & i \\ i & 1 \end{bmatrix},$$

the input transforms as:

$$a_1^\dagger \rightarrow \frac{1}{\sqrt{2}}(a_3^\dagger + ia_4^\dagger), \tag{27}$$

and the two-photon state becomes:

$$|\Psi\rangle \rightarrow \frac{1}{2\sqrt{2}}(a_3^\dagger + ia_4^\dagger)^2 |\text{vac}\rangle. \tag{28}$$

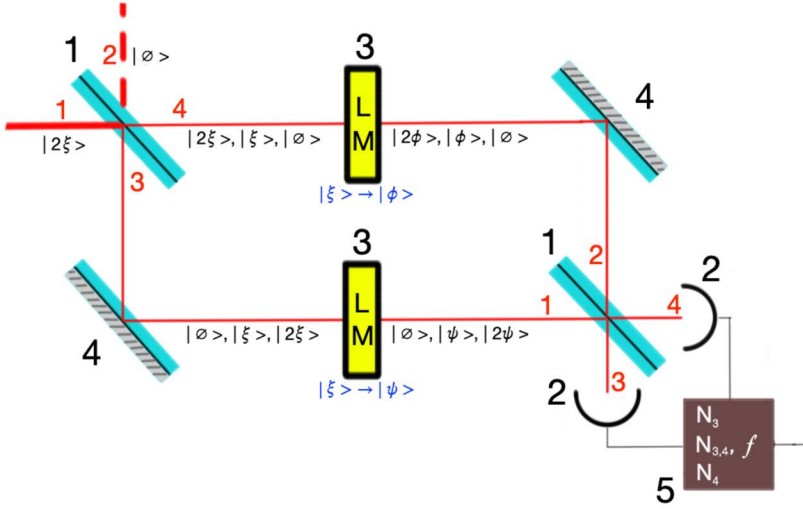

**Fig 1. Mach–Zehnder interferometer with a light modulator on each arm.** The occupation number options for the two photons at different stages are shown. The interferometer includes: (1) two 50:50 beam splitters with input ports (red: 1,2) and output ports (red: 3,4); (2) two photon-number resolving detectors; (3) two light modulators implementing $|\eta\rangle \to |\phi\rangle$ and $|\eta\rangle \to |\psi\rangle$; (4) two mirrors; (5) a coincidence counter detecting: (*i*) two photons at port 3, (*ii*) two at port 4, or (*iii*) one at each port. This counter determines the output function **f**.

After modulation by two light modulators:

$$a_3^\dagger \to b_1^\dagger, \qquad a_4^\dagger \to c_2^\dagger, \tag{29}$$

with

$$b_i^\dagger |\text{vac}\rangle = |\phi\rangle_i, \qquad c_i^\dagger |\text{vac}\rangle = |\psi\rangle_i, \tag{30}$$

the commutation rules are:

$$[b_i, c_j] = 0, \quad [b_i, c_j^\dagger] = \delta_{ij} \langle\phi|\psi\rangle. \tag{31}$$

After the first beam splitter and modulation, the state becomes:

$$|\Psi\rangle \to \frac{1}{2\sqrt{2}}(b_1^\dagger + ic_2^\dagger)^2 |\text{vac}\rangle. \tag{32}$$

### 3.1.1 Passage through the second beam splitter.

The second symmetric beam splitter acts as:

$$b_1^\dagger \to \frac{1}{\sqrt{2}}(b_3^\dagger + ib_4^\dagger), \qquad c_2^\dagger \to \frac{1}{\sqrt{2}}(ic_3^\dagger + c_4^\dagger), \tag{33}$$

leading to:

$$\begin{aligned}|\Psi\rangle \to \frac{1}{4\sqrt{2}} \big(& b_3^\dagger b_3^\dagger - b_4^\dagger b_4^\dagger + c_3^\dagger c_3^\dagger - c_4^\dagger c_4^\dagger \\ &+ 2ib_3^\dagger b_4^\dagger - 2b_3^\dagger c_3^\dagger + 2ib_3^\dagger c_4^\dagger - 2ib_4^\dagger c_3^\dagger - 2b_4^\dagger c_4^\dagger - 2ic_3^\dagger c_4^\dagger \big) |\text{vac}\rangle. \end{aligned} \tag{34}$$

### 3.1.2 Orthogonal basis for $|\psi\rangle$.

To simplify the evaluation of detection probabilities, we define the normalized state orthogonal to $|\phi\rangle$:

$$|\bar{\phi}\rangle = \frac{|\psi\rangle - \langle\phi|\psi\rangle\,|\phi\rangle}{\sqrt{1 - |\langle\phi|\psi\rangle|^2}},$$

(35)

with

$$\langle\bar{\phi}|\phi\rangle = 0, \qquad \langle\psi|\bar{\phi}\rangle = \sqrt{1 - |\langle\phi|\psi\rangle|^2}.$$

(36)

Introducing creation/annihilation operators $d_i^\dagger$, $d_i$ for $|\bar{\phi}\rangle$, the commutation rules are:

$$[d_i, b_j] = [d_i, c_j] = 0,$$
$$[d_i, b_j^\dagger] = 0, \qquad [d_i, c_j^\dagger] = \delta_{ij}\sqrt{1 - |\langle\phi|\psi\rangle|^2}.$$

(37)

### 3.1.3 Detection probabilities.

We now define the output probabilities:

- Coincidence:

$$P_{3,4} = \sum_{x,y\in\{b,d\}} \left|\langle\text{vac}|\,x_3 y_4\,|\Psi\rangle\right|^2.$$

(38)

- Bunching at detector 3:

$$P_3 = \left|\langle\text{vac}|\frac{b_3 b_3}{\sqrt{2}}\,|\Psi\rangle\right|^2 + \left|\langle\text{vac}|\,b_3 d_3\,|\Psi\rangle\right|^2 + \left|\langle\text{vac}|\frac{d_3 d_3}{\sqrt{2}}\,|\Psi\rangle\right|^2.$$

(39)

- Bunching at detector 4:

$$P_4 = \left|\langle\text{vac}|\frac{b_4 b_4}{\sqrt{2}}\,|\Psi\rangle\right|^2 + \left|\langle\text{vac}|\,b_4 d_4\,|\Psi\rangle\right|^2 + \left|\langle\text{vac}|\frac{d_4 d_4}{\sqrt{2}}\,|\Psi\rangle\right|^2.$$

(40)

Using the expressions above and computing the inner products, we obtain the final results:

$$P_{3,4} = \frac{1}{2}\left(1 - \text{Re}\{\langle\phi|\psi\rangle\}^2\right),$$

(41)

$$P_3 = \frac{1}{4}\left(1 - \text{Re}\{\langle\phi|\psi\rangle\}\right)^2,$$

(42)

$$P_4 = \frac{1}{4}\left(1 + \text{Re}\{\langle\phi|\psi\rangle\}\right)^2,$$

(43)

which satisfy the normalization:

$$P_{3,4} + P_3 + P_4 = 1. \tag{44}$$

### 3.1.4 Recovering the scalar product from measurements.

From (42) and (43), we can directly extract

$$\text{Re}\{\langle\phi|\psi\rangle\} = P_4 - P_3. \tag{45}$$

To retrieve $\text{Im}\{\langle\phi|\psi\rangle\}$, we perform a second measurement session by modifying the modulators as follows:

$$b_i^\dagger |\text{vac}\rangle = i|\phi\rangle_i, \qquad c_i^\dagger |\text{vac}\rangle = |\psi\rangle_i, \tag{46}$$

and define $P'_{3,4}$, $P'_3$ and $P'_4$ accordingly. In this case,

$$\text{Im}\{\langle\phi|\psi\rangle\} = P'_4 - P'_3. \tag{47}$$

Thus, the scalar product can be reconstructed solely from detection statistics:

$$\langle\phi|\psi\rangle = (P_4 - P_3) + i(P'_4 - P'_3), \tag{48}$$

while for its modulus:

$$\left|\langle\phi|\psi\rangle\right|^2 = 2\left(1 - P_{3,4} - P'_{3,4}\right) = (P_4 - P_3)^2 + (P'_4 - P'_3)^2. \tag{49}$$

With reference to the pre-activation function $\mathcal{T}_2$ defined in (25), we obtain

$$\mathcal{T}_2 = \text{Re}\,\langle\phi|\psi\rangle\cos\theta + \text{Im}\,\langle\phi|\psi\rangle\sin\theta + b = (P_4 - P_3)\cos\theta + (P'_4 - P'_3)\sin\theta + b. \tag{50}$$

With an MZ interferometer, we can also efficiently evaluate $\mathcal{T}_1$, which in this case reads:

$$\mathcal{T}_1 = \left|\langle\phi|\psi\rangle\right|^2 + b = 2\left(1 - P_{3,4} - P'_{3,4}\right) + b = (P_4 - P_3)^2 + (P'_4 - P'_3)^2 + b. \tag{51}$$

### 3.1.5 Alternative setups.

If both beam splitters are replaced with dielectric 50:50 beam splitters, described by the transfer matrix

$$\frac{1}{\sqrt{2}}\begin{bmatrix} 1 & 1 \\ 1 & -1 \end{bmatrix},$$

all formulas remain unchanged. Conversely, if only one of the two is replaced with a dielectric beam splitter, the real part in Eqs. (41)–(43) is replaced by the imaginary part. This yields an alternative procedure for retrieving $\text{Im}\{\langle\phi|\psi\rangle\}$ in the second measurement session, without the need to modify the light modulators as in (46).

### 3.1.6 Decoherence and noise effects.

Observations of phenomena in the quantum domain are generally affected by decoherence [36–38]. Decoherence arises from the interaction between a quantum system and its environment, or equivalently from the transfer of information from the system to the environment [37]. As a consequence, phase coherence in a superposition of physical states associated with definite outcomes of a given observable progressively degrades, and the inherent quantum indeterminacy is effectively converted into the classical indeterminacy of statistical

mixtures [39–42]. From a mathematical perspective, this transition is described by an exponential decay in time of the off-diagonal elements of the system density matrix.

In the MZ interferometer setup depicted in Fig 1, such off-diagonal elements give rise, in the probabilities (41), (42), and (43), to terms proportional to $\text{Re}\{\langle\phi|\psi\rangle\}$ and $\text{Re}\{\langle\phi|\psi\rangle\}^2$. These contributions are genuinely quantum in nature, originating from the which-path indeterminacy of the photons inside the interferometer and having no classical counterpart. Decoherence attenuates these quantum interference terms through a multiplicative factor $V$, commonly referred to as the visibility of the interferometer [43]:

$$P_{3,4} = \frac{1}{2}\left(1 - V^2\text{Re}\{\langle\phi|\psi\rangle\}^2\right),$$
(52)

$$P_3 = \frac{1}{4}\left(1 - V\text{Re}\{\langle\phi|\psi\rangle\}\right)^2,$$
(53)

$$P_4 = \frac{1}{4}\left(1 + V\text{Re}\{\langle\phi|\psi\rangle\}\right)^2.$$
(54)

Visibility takes values in the interval $[0, 1]$: values of $V$ close to unity correspond to nearly ideal constructive and destructive quantum interference, while values approaching zero indicate the onset of classical behavior. The visibility of a given interferometer can be experimentally estimated through preliminary calibration measurements.

The probabilities $P_{3,4}$, $P_3$, and $P_4$ in Eqs. (52)–(54) remain properly normalized, and their relationship with the real part of the scalar product $\langle\phi|\psi\rangle$ is given by

$$\text{Re}\{\langle\phi|\psi\rangle\} = \frac{P_4 - P_3}{V}.$$
(55)

An analogous relation holds for the imaginary part of the scalar product, which can be reconstructed from the probabilities $P'_3$ and $P'_4$ measured in the complementary configuration:

$$\text{Im}\{\langle\phi|\psi\rangle\} = \frac{P'_4 - P'_3}{V}.$$
(56)

Quantum measurements are also affected by noise, often at a more intrinsic level than classical measurements, since quantum phenomena are inherently probabilistic. In addition to statistical errors arising from finite sampling of probabilities by event occurrences, other relevant noise sources include dark counts—i.e., detector clicks not originating from photons produced by the interferometer—finite detector efficiency, and the possible lack of photon-number-resolving capabilities. Moreover, environmental fluctuations can induce variability in the interferometer visibility $V$.

By replacing probabilities with the experimentally observed detection frequencies $\nu_i$ and $\nu'_i$, Eqs. (55) and (56) become

$$\frac{\nu_4 - \nu_3}{V} = \text{Re}\{\langle\phi|\psi\rangle\} + \epsilon_R, \quad \frac{\nu'_4 - \nu'_3}{V} = \text{Im}\{\langle\phi|\psi\rangle\} + \epsilon_I,$$
(57)

where $\epsilon_i$, $\epsilon'_i$, and $\epsilon_V$ denote the fluctuations of $\nu_i$, $\nu'_i$, and $V$ around their mean values. To first order, these uncertainties can be approximated as

$$\epsilon_R \simeq \frac{1}{V}\left(\epsilon_3 + \epsilon_4 + \text{Re}\{\langle\phi|\psi\rangle\}\epsilon_V\right), \quad \epsilon_I \simeq \frac{1}{V}\left(\epsilon_3' + \epsilon_4' + \text{Im}\{\langle\phi|\psi\rangle\}\epsilon_V\right).$$

(58)

Using Eq. (23), an analogous expression can be derived for the HOM interferometer:

$$\frac{1 - 2\nu_{coinc.}}{V} = \left|\langle\phi|\psi\rangle\right|^2 + \epsilon_M,$$

(59)

where

$$\epsilon_M \simeq \frac{1}{V}\left(2\epsilon_{coinc.} + \left|\langle\phi|\psi\rangle\right|^2 \epsilon_V\right).$$

(60)

## 3.2 Computational cost of QONs

In this section, we analyze the computational costs of the output functions and their gradients for the QONs previously introduced. For simplicity, we consider the case of an ideal QON. From (3), (4), (19), (23), (45), (47), the output functions read:

$$f_1 = \begin{cases} h\left(1 - 2P_{\text{coinc.}} + b\right) & \text{for HOM,} \\ h\left(2(1 - P_{3,4} - P_{3,4}') + b\right) & \text{for MZ,} \end{cases}$$

$$f_2 = h\left((P_4 - P_3)\cos\theta + (P_4' - P_3')\sin\theta + b\right) \text{ only for MZ.}$$

In all these cases, the computational cost during the inference phase, i.e., when querying a trained QON, is independent of $N$, the number of parameters. This results in a super-exponential speed-up compared to the output function of a classical neuron, whose evaluation scales as $\mathcal{O}(N)$.

Concerning the computational costs of an artificial neuron during training, which mainly involve computing the derivatives of the output functions $f_1$ and $f_2$, they must be analyzed according to the chosen photon modulation scheme.

**3.2.1 Amplitude/Intensity modulation.** In the case of amplitude modulation, from (6), (11), and (12) the gradients $\frac{\partial f_1}{\partial \lambda_k}$ read:

$$\frac{\partial}{\partial \lambda_k} f_1\left(\langle\phi|\psi\rangle\right) = h'\left(\|\langle\phi|\psi\rangle\|^2 + b\right)\sqrt{\|\langle\phi|\psi\rangle\|^2}\left[\frac{\mu_k}{\|\boldsymbol{\mu}\|\,\|\boldsymbol{\lambda}\|} - \frac{\lambda_k}{\|\boldsymbol{\lambda}\|^2}\sqrt{\|\langle\phi|\psi\rangle\|^2}\right],$$

(61)

so that, by (23):

$$\frac{\partial}{\partial \lambda_k} f_1\left(\langle\phi|\psi\rangle\right) = h'\left(1 - 2P_{\text{coinc.}} + b\right)\sqrt{1 - 2P_{\text{coinc.}}}\left[\frac{\mu_k}{\|\boldsymbol{\mu}\|\,\|\boldsymbol{\lambda}\|} - \frac{\lambda_k}{\|\boldsymbol{\lambda}\|^2}\sqrt{1 - 2P_{\text{coinc.}}}\right].$$

(62)

Although the term $\langle\phi|\psi\rangle$ is estimated by $P_{\text{coinc.}}$, the computational cost of $\frac{\partial f_1}{\partial \lambda_k}$ remains $\mathcal{O}(N)$. This is because determining $\|\boldsymbol{\mu}\|$ and $\|\boldsymbol{\lambda}\|$ both require $\mathcal{O}(N)$ computational resources. Since these calculations are common to all $N$ components of the gradient, the overall computational cost is still $\mathcal{O}(N)$. Thus, when it comes to calculating the gradients, QONs do not have any advantages over classical neurons. A similar analysis applies to the intensity modulation case.

**3.2.2 Phase modulation.** From (7) and (20), the partial derivatives of $f_1$ are given by:

$$\frac{\partial}{\partial \lambda_k} f_1\left(\langle\phi|\psi\rangle\right) = -\frac{2}{N}h'\left(|\langle\phi|\psi\rangle|^2 + b\right) \text{Im}\left\{\langle\phi|\psi\rangle\, e^{i(\lambda_k - \mu_k)}\right\}$$

$$= -\frac{2}{N}h'\left(|\langle\phi|\psi\rangle|^2 + b\right)\left[\text{Re}\left\{\langle\phi|\psi\rangle\right\}\sin\left(\lambda_k - \mu_k\right) + \text{Im}\left\{\langle\phi|\psi\rangle\right\}\cos\left(\lambda_k - \mu_k\right)\right]. \tag{63}$$

Using the HOM interference measurement results from (23), Eq. (63) simplifies to:

$$\frac{\partial}{\partial \lambda_k} f_{1,HOM}\left(\langle\phi|\psi\rangle\right) = -\frac{2}{N}h'\left(1 - 2P_{\text{coinc.}} + b\right)\text{Im}\left\{\langle\phi|\psi\rangle\, e^{i(\lambda_k - \mu_k)}\right\}. \tag{64}$$

The computational cost of evaluating this derivative is $\mathcal{O}(N)$, since it requires explicitly computing $\langle\phi|\psi\rangle$, which cannot be derived directly from its modulus $\sqrt{|\langle\phi|\psi\rangle|^2}$. However, because $\langle\phi|\psi\rangle$ is a common term for all first-order derivatives, the total cost for computing the gradient $\frac{\partial f_1}{\partial \boldsymbol{\lambda}}$ remains $\mathcal{O}(N)$, matching that of a classical neuron.

Alternatively, using the MZ interference measurement results from (45), (47), and (49), Eq. (63) becomes:

$$\frac{\partial}{\partial \lambda_k} f_{1,MZ}\left(\langle\phi|\psi\rangle\right) = -\frac{2}{N}h'\left[2\left(1 - P_{3,4} - P'_{3,4}\right) + b\right] \cdot$$

$$\cdot\left[(P_4 - P_3)\sin\left(\lambda_k - \mu_k\right) + \left(P'_4 - P'_3\right)\cos\left(\lambda_k - \mu_k\right)\right]. \tag{65}$$

In this case, the computational cost of each derivative $\frac{\partial}{\partial \lambda_k} f_1$ is constant, i.e., independent of $N$. Nevertheless, since $N$ derivatives must be computed, the overall time complexity for the full gradient $\frac{\partial f_1}{\partial \boldsymbol{\lambda}}$ remains $\mathcal{O}(N)$.

A similar analysis applies to $f_2$. Using (7) and (21), its derivatives are given by:

$$\frac{\partial}{\partial \lambda_k} f_2\left(\langle\phi|\psi\rangle\right) = \frac{1}{N}h'\left(\text{Re}\,\langle\phi|\psi\rangle\cos\theta + \text{Im}\,\langle\phi|\psi\rangle\sin\theta + b\right)\sin(\mu_k - \lambda_k + \theta),$$

which, using the MZ measurement results from (45) and (47), becomes:

$$\frac{\partial}{\partial \lambda_k} f_2\left(\langle\phi|\psi\rangle\right) = \frac{1}{N}h'\left[(P_4 - P_3)\cos\theta + \left(P'_4 - P'_3\right)\sin\theta + b\right]\sin(\mu_k - \lambda_k + \theta).$$

In conclusion, training a QON under phase modulation can require significantly less computation per derivative compared to a classical neuron, especially when using MZ interferometry, since the quantities of interest are extracted directly from detection statistics. However, as the gradient has $N$ components, the overall time complexity for backpropagation remains $\mathcal{O}(N)$.

## 3.3 Quantum optical neural network architecture

We now extend the QON model to define a multilayer quantum optical neural network. Each layer is indexed by $l = 1, \ldots, L$, and contains $N_l$ neurons. A neuron in position $p$ of layer $l$ is denoted by the pair $(l; p)$.

Each neuron is characterized by a set of parameters $\boldsymbol{\lambda}_{l;p} = (\lambda_{l;p,1}, \ldots, \lambda_{l;p,N_{l-1}})$, from which the complex weight coefficients $\boldsymbol{w}_{l;p} = (w_{l;p,1}, \ldots, w_{l;p,N_{l-1}})$ are derived. The inputs to each neuron are denoted by $\boldsymbol{v}_l = (v_{l;1}, \ldots, v_{l;N_{l-1}})$, shared across neurons in the same layer, while the outputs of layer $l$ are collected into the vector $\boldsymbol{f}_l = (f_{l;1}, \ldots, f_{l;N_l})$.

The quantum states interfering at the node $(l; p)$ are $|\phi_{l;p}\rangle$ and $|\psi_l\rangle$, and the output of each neuron is given by a function $f$ of their scalar product. For $l \geq 2$, the dependencies are as follows:

$$w_{l;p} \equiv w_{l;p}(\lambda_{l;p}),$$
$$v_l \equiv v_l(f_{l-1}),$$
$$f_{l;p} = f\left(\sum_{j=1}^{N_{l-1}} w_{l;p,j}^* v_{l;j}\right) = f\left(\langle \phi_{l;p} | \psi_l \rangle\right). \tag{66}$$

**3.3.1 Gradient computation for training.** To apply gradient descent during training, we compute the derivative of the final output $f_{L;p}$ with respect to a generic parameter $\lambda_{\bar{l};\bar{p};\bar{j}}$. From (66), we obtain:

$$\frac{\partial f_{L;p}}{\partial \lambda_{\bar{l};\bar{p};\bar{j}}} = f'\left(\langle \phi_{L;p} | \psi_L \rangle\right) \sum_{j=1}^{N_{L-1}} w_{L;p,j}^* \sum_{k=1}^{N_{L-1}} \frac{\partial v_{L;j}}{\partial f_{L-1;k}} \frac{\partial f_{L-1;k}}{\partial \lambda_{\bar{l};\bar{p};\bar{j}}}. \tag{67}$$

This recursive formulation can be conveniently expressed in matrix form:

$$\frac{\partial \boldsymbol{f}_L^T}{\partial \lambda_{\bar{l};\bar{p};\bar{j}}} = \text{diag}\left\{f'\left(\langle \phi_L | \psi_L \rangle\right)\right\} \hat{w}_L^* \frac{\partial \boldsymbol{v}_L^T}{\partial \boldsymbol{f}_{L-1}} \frac{\partial \boldsymbol{f}_{L-1}^T}{\partial \lambda_{\bar{l};\bar{p};\bar{j}}}, \tag{68}$$

where:

- $\hat{w}_L = \left(w_{L;p,j}\right)_{1 \le p \le N_L,\, 1 \le j \le N_{L-1}}$ is the weight matrix;
- $\frac{\partial \boldsymbol{v}_L^T}{\partial \boldsymbol{f}_{L-1}} = \left(\frac{\partial v_{L;j}}{\partial f_{L-1;k}}\right)_{1 \le j,k \le N_{L-1}}$;
- $\frac{\partial \boldsymbol{f}_l^T}{\partial \lambda_{\bar{l};\bar{p};\bar{j}}} = \left(\frac{\partial f_{l;1}}{\partial \lambda_{\bar{l};\bar{p};\bar{j}}}, \ldots, \frac{\partial f_{l;N_l}}{\partial \lambda_{\bar{l};\bar{p};\bar{j}}}\right)^T$;
- $\text{diag}\left\{f'\left(\langle \phi_L | \psi_L \rangle\right)\right\}$ is a diagonal matrix with entries $f'(\langle \phi_{L;p} | \psi_L \rangle)$ on the diagonal.

By recursively applying (68), we obtain the general formula:

$$\frac{\partial \boldsymbol{f}_L^T}{\partial \lambda_{\bar{l};\bar{p};\bar{j}}} = \left[\overset{\curvearrowleft L}{\prod_{l=\bar{l}+1}} \text{diag}\left\{f'\left(\langle \phi_l | \psi_l \rangle\right)\right\} \hat{w}_l^* \frac{\partial \boldsymbol{v}_l^T}{\partial \boldsymbol{f}_{l-1}}\right] \text{diag}\left\{f'\left(\langle \phi_{\bar{l}} | \psi_{\bar{l}} \rangle\right)\right\} \frac{\partial \hat{w}_{\bar{l}}^*}{\partial \lambda_{\bar{l};\bar{p};\bar{j}}}, \tag{69}$$

where $\overset{\curvearrowleft L}{\prod_{l=\bar{l}+1}}$ indicates the product of a sequence in which the factors with higher indices are positioned to the left of those with lower indices, while

$$\frac{\partial \hat{w}_{\bar{l}}}{\partial \lambda_{\bar{l};\bar{p};\bar{j}}} = \left(\frac{\partial w_{\bar{l};j,k}}{\partial \lambda_{\bar{l};\bar{p};\bar{j}}}\right)_{1 \le j,k \le N_{\bar{l}}}.$$

**3.3.2 Physical setup and beam-splitter configuration.** Fig 2 illustrates a schematic implementation of a feed-forward, fully-connected QONN based on MZ interferometers. Other network architectures, such as recurrent or convolutional neural networks, can also be considered, potentially replacing MZ with HOM interferometers. Each neuron (one highlighted in red) receives light from a common coherent laser source (bottom left), which is distributed by a cascade of beam splitters. The laser intensity is adjusted so that each QON receives at most a pair of photons at a time—identical due to the coherent nature of the laser beam—as required in Section 3.1. Inputs (bottom right) and weights (left) are encoded into quantum states via light modulators.

In the first layer, the right-side modulators are driven by input data, while the left-side modulators encode the weights. In subsequent layers, the inputs to each neuron correspond to the outputs of QONs from the previous layer. All outputs are also routed to the weight update module for gradient estimation.

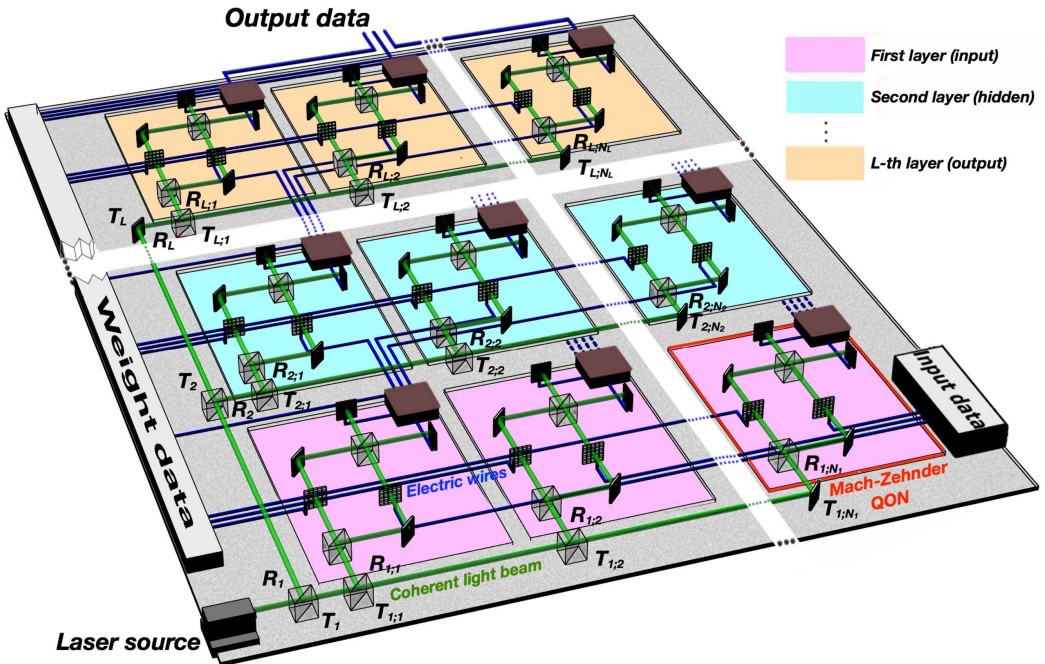

**Fig 2. Schematic architecture of a QONN with *L* layers implemented using Mach–Zehnder QONs.** Blue lines indicate electrical connections; green lines indicate laser paths. Each layer is color-coded. In the first layer (pink), the right-side light modulators are governed by input data, while weights govern the left-side light modulators. In deeper layers, right-side light modulators are driven by the outputs of the previous layer. QON outputs also feed the weight module for training. For clarity, not all connections are shown.

Beam splitters are organized at two levels: global ($R_l$, $T_l$), distributing light across layers, and local ($R_{l;p}$, $T_{l;p}$), distributing light to individual neurons within each layer. To ensure proper synchronization, all neurons must receive the same photon flux $F$. Let

$$N_{\text{tot}} = \sum_{l=1}^{L} N_l,$$

be the total number of neurons, and $F_{\text{tot}}$ the total flux from the laser. Then, the flux $F_l$ entering the *l*-th layer must satisfy:

$$F_l = \frac{N_l}{N_{\text{tot}}} F_{\text{tot}}. \tag{70}$$

To fulfill condition (70), the global reflection coefficients $R_l$ must be set as:

$$R_l = \begin{cases} 1 - \dfrac{N_1}{N_{\text{tot}}}, & \text{if } l = 1, \\[2mm] \dfrac{N_l}{\sum_{l'=1}^{L} N_{l'}}, & \text{for } 1 < l \leq L, \end{cases} \qquad T_l = 1 - R_l. \tag{71}$$

Similarly, within each layer, the local beam splitters must satisfy:

$$R_{l;p} = \frac{1}{N_l - p + 1}, \qquad T_{l;p} = 1 - R_{l;p}.$$

(72)

Note that $R_L = 1$ and $R_{l;N_l} = 1$, ensuring that the last layer and the last neuron within each layer receive all the remaining light.

## 4 Practical implementation of a QON

Although practical, implementing a QON presents several technical challenges. In particular, the use of spatial light modulators and spatially resolved detectors typically requires the optical setup to be realized almost entirely in free space. Moreover, the highest-performing SLMs—both in phase and amplitude modulation—are usually reflective architectures [44], which are characterized by limited acceptance angles. This constraint significantly complicates the design of interferometric configurations when compared to schemes based on transmissive SLMs, such as the idealized MZ setup depicted in Fig 1 or the HOM-based configuration described in [25]. Reflective geometries generally require additional optical components, including beam splitters, mirrors, and relay optics, to properly direct and recombine optical paths. As a consequence, careful optical design and precise alignment are necessary to maximize interference visibility.

In addition, in the MZ-based QON, the reconstruction of the bunching probabilities $P_3$ and $P_4$ appearing in Eqs. (45) and (47) requires photon-number-resolving detectors [45], which are more complex than the non-resolving detectors sufficient for the HOM-based QON.

Beyond physical feasibility, practical deployment of QONs as artificial neurons also requires sufficiently short processing times. In Section 5, we perform simulations assuming an event-counting time of *1* s. Under typical laboratory conditions—considering factors such as laser pulse repetition rate and detector efficiency—shorter observation times may lead to significant statistical deviations between measured event frequencies and their underlying probabilities. Preliminary simulations indicate that QON performance degrades markedly at high noise levels. However, an observation time of *1* s per scalar product would be impractical for an artificial neuron, particularly during training.

To mitigate this limitation, careful optimization of the optical apparatus is required, including the use of high-efficiency detectors, low-loss beam splitters, and stable optical components to minimize photon losses. In addition, operating at higher laser repetition rates can provide sufficient statistics within much shorter acquisition times. For instance, by employing superconducting nanowire single-photon detectors [46] with efficiencies of approximately 0.9 and photon emission rates on the order of *1* MHz—rather than single-photon avalanche diode (SPAD) detectors [47] with efficiencies around 0.4 and emission rates of $\sim 7 \times 10^4$ Hz, as assumed in the simulations of Section 5—the required observation time could be reduced to approximately $10^{-2}$ s. Finally, as indicated by Eqs. (58) and (60), maintaining high and stable visibility, achievable through effective environmental isolation of the optical setup, is a critical requirement for reliable QON operation.

## 5 Benchmarking QON architectures: Performance of pre-activation variants

We introduced different implementations of quantum optical neurons, which vary both in the experimental setup (Mach–Zehnder or Hong–Ou–Mandel) and in the type of photon modulation (phase, amplitude, or intensity). While the modulation strategy determines how input $\mu$ and weight parameters $\lambda$ contribute to the scalar product $\langle \phi | \psi \rangle$, the interferometric architecture dictates how this scalar product is transformed into a pre-activation.

These choices result in different mathematical forms of the pre-activation function $\mathcal{T}(\mu, \lambda)$, which acts as the core computation within the QON. Below, we summarize the quantum-inspired variants explored in this study:

1. MZ interferometer with phase modulation (from (18) and (25) with $A = 1$):

$$\mathcal{T}_{2,\theta} = \frac{1}{N} \sum_{j=1}^{N} \cos(\mu_j - \lambda_j + \theta) + b; \tag{73}$$

2. HOM (or MZ) interferometer with phase modulation (from (17) and (22)):

$$\mathcal{T}_{1,\text{ph.}} = \frac{1}{N^2} \left| \sum_{j=1}^{N} e^{i(\mu_j - \lambda_j)} \right|^2 + b; \tag{74}$$

3. HOM interferometer with amplitude modulation (from (11) and (22)):

$$\mathcal{T}_{1,\text{am.}} = \left( \sum_{j=1}^{N} \frac{\mu_j \lambda_j}{\|\boldsymbol{\mu}\| \|\boldsymbol{\lambda}\|} \right)^2 + b; \tag{75}$$

4. HOM interferometer with intensity modulation (from (13) and (22)):

$$\mathcal{T}_{1,\text{in.}} = \left( \sum_{j=1}^{N} \sqrt{\frac{\mu_j \lambda_j}{|\boldsymbol{\mu}|_1 |\boldsymbol{\lambda}|_1}} \right)^2 + b. \tag{76}$$

In each case, $b$ is a bias term, while $\theta$ in (73) is a tunable phase offset (we considered $\theta = 0$ and $\pi/4$). These functions can be compared to the standard linear pre-activation of a classical artificial neuron:

$$\mathcal{T}_{\text{classical}} = \sum_{j=1}^{N} \mu_j \lambda_j + b. \tag{77}$$

We consider both the ideal formulation of each QON variant and a non-ideal version incorporating decoherence and noise effects, as described in Section 3.1.6. This allows assessing not only accuracy and convergence, but also the robustness of quantum-inspired pre-activations under realistic imperfections.

As discussed in Section 3.1.6, decoherence and noise effects induce fluctuations in the estimation of the real and imaginary parts, or of the squared modulus, of the scalar product between two photon quantum states, as inferred from the interference patterns of an MZ or HOM interferometer, respectively. Since the pre-activation functions $\mathcal{T}$ defined in Eqs. (73)–(76) depend explicitly on these quantities (apart from the bias term $b$), their evaluation through a QON is inevitably affected by such fluctuations.

From Eqs. (58) and (60), it follows that the uncertainty in $\mathcal{T}$ arises from two main contributions: fluctuations in the interferometer visibility and statistical deviations of the observed event frequencies from the corresponding probabilities, both of which depend on the mean visibility value. For the HOM interferometer, Tsujimoto et al. [48] reports a visibility of $V_{\text{HOM}} \simeq 0.87 \pm 0.04$. For the MZ interferometer, Kim et al. [49] reports $V_{\text{MZ}}^2 \simeq 0.75$ for coincidence events, but does not provide an uncertainty estimate. We therefore assume the same relative uncertainty as in the HOM case and set $V_{\text{MZ}} \simeq 0.85 \pm 0.04$.

Under these assumptions, Eqs. (58) and (60) allow us to model visibility-induced fluctuations of $\mathcal{T}$ as zero-mean Gaussian noise with standard deviation $\delta\mathcal{T}_V \approx 0.046\,\mathcal{T}$ for the HOM interferometer and $\delta\mathcal{T}_V \approx 0.047\,\mathcal{T}$ for the MZ interferometer.

Additional fluctuations arise from finite sampling statistics and detector imperfections, and depend on the measurement apparatus and observation time. Assuming a photon emission rate of $7 \times 10^4$ Hz, detector efficiency of 0.4 for typical SPAD detectors, dead time of $10^{-5}$ s, and dark count rate of $3 \times 10^2$ Hz per detector, preliminary simulations indicate that, for an observation time of *1* s, these effects can be approximated by zero-mean Gaussian noise with standard deviation $\delta\mathcal{T}_{\text{coinc.}} \approx 0.02(1 - \mathcal{T})$ for the HOM interferometer and $\delta\mathcal{T}_{\text{bunch}} \approx 0.01\sqrt{1 + \mathcal{T}^2}$ for the MZ interferometer.

By assuming these two noise sources to be independent, the output pre-activation of a realistic QON can be modeled as a Gaussian random variable with mean value $\mathcal{T}$ given by Eqs. (73)–(76) and standard deviation

$$\delta\mathcal{T} = \sqrt{0.02^2 \left(1 - \mathcal{T}\right)^2 + 0.046^2 \, \mathcal{T}^2},$$

for the HOM-based QONs, and

$$\delta\mathcal{T} = \sqrt{0.01^2 \left(1 + \mathcal{T}^2\right) + 0.047^2 \, \mathcal{T}^2},$$

for the MZ-based QONs, for both $\theta = 0$ and $\theta = \pi/4$.

In the remainder of this section, we experimentally evaluate the six pre-activation functions above—five quantum-inspired and one classical—using two types of classification tasks: (*i*) binary classification using a single QON, and (*ii*) multiclass classification using a full feed-forward network composed of QON layers. The goal is to investigate how the design of the pre-activation function affects learning dynamics, accuracy, convergence, and robustness across tasks of increasing complexity.

It is worth noting that all experiments presented in this study were conducted using software-based simulations of QONs implemented in PyTorch. Each quantum optical model was translated into a differentiable pre-activation function that faithfully reflects the mathematical formulation derived from its physical interpretation. However, no physical optical hardware was used: the aim was to evaluate the computational potential of these physically inspired neurons within standard learning frameworks.

## 5.1 Binary classification with a single neuron

To evaluate the computational effectiveness of different quantum-inspired pre-activations $\mathcal{T}$, we conducted controlled experiments using a deliberately simple setup: a single QON trained end-to-end on a binary classification task.

We used two well-known image recognition datasets: MNIST [50] and FashionMNIST [51], each restricted to a binary setting with class labels 0 and 1. For MNIST, this corresponds to handwritten digits 0 and 1, while for FashionMNIST, the labels refer to T-shirts/tops and trousers, respectively. Each $28 \times 28$ grayscale image was flattened into a vector $\boldsymbol{\mu} \in \mathbb{R}^{784}$ and normalized to lie in [0, 1]. This basic preprocessing was sufficient, as normalization specific to each QON variant was embedded in their respective formulations.

The QON receives the input $\boldsymbol{\mu}$ and computes a scalar pre-activation $\mathcal{T}(\boldsymbol{\mu}, \boldsymbol{\lambda})$ using a learnable weight vector $\boldsymbol{\lambda}$ of matching dimension. For the classical neuron, we include the additive bias term. In contrast, we found that including the bias term in quantum-inspired pre-activations consistently degraded performance. We hypothesize that this behavior is primarily related to optimization difficulties rather than to a fundamental limitation of quantum optical neurons. QONs encode information through highly nonlinear pre-activation functions that often rely on normalized or bounded interference-based quantities, reflecting relative and coherent relationships between optical fields. In this context, the introduction of a purely linear and additive bias term may disrupt the intrinsic normalization underlying the interference process, effectively altering the geometry of the loss landscape. Such a mismatch can hinder gradient propagation, leading to less stable convergence and weaker generalization. Based on these empirical observations, we excluded the bias

term from all QON variants and retained it only for the classical baseline, as this choice consistently improved training stability while preserving the physical interpretability of the optical analogy. Nevertheless, this aspect will be the subject of future investigation to develop a bias formulation that could further enhance QON performance.

All scalar pre-activations were followed by batch normalization [52] and a final sigmoid activation to output a probability. Batch normalization was consistently found to improve stability and convergence across all models.

We implemented six differentiable PyTorch modules, each corresponding to a different pre-activation function, as defined above. These include the classical inner product and five quantum-inspired functions derived from Mach–Zehnder or Hong–Ou–Mandel interferometry, using various encoding schemes (phase, amplitude, intensity). For MZ-based neurons, we tested both the vanilla cosine formulation and a phase-shifted variant using $\theta = \pi/4$.

All models were trained using the Adam optimizer [53] with a learning rate of 0.01, batch size 64, and binary cross-entropy loss, for a total of 25 epochs. To assess the robustness of each pre-activation variant, every experiment was repeated five times independently under identical training conditions. All reported curves correspond to the mean across runs, while the shaded regions denote the standard deviation.

Under ideal conditions, most models converged rapidly and achieved excellent generalization on MNIST (Fig 3, top row), with final test accuracies close to or above 99%. MZ-based neurons (both cosine and phase-shifted) displayed the most stable training dynamics and the smallest variance bands. HOM-based neurons showed more heterogeneous behavior: the intensity-based encoding converged more slowly, while the amplitude-modulated HOM neuron consistently underperformed relative to the other QON variants.

In the non-ideal setting (Fig 3, bottom row), the impact of noise and decoherence became evident. MZ neurons remained comparatively robust, maintaining stable convergence profiles. HOM-based neurons were significantly more affected: phase and intensity variants showed increased variance and reduced accuracy, reflecting their greater sensitivity to perturbations in the underlying interference patterns.

The FashionMNIST results in Fig 4 confirm the same qualitative trends but with amplified differences among variants. In particular, HOM phase modulation exhibited sharp oscillations and larger uncertainty bands in the non-ideal scenario, whereas MZ neurons preserved stable generalization across runs.

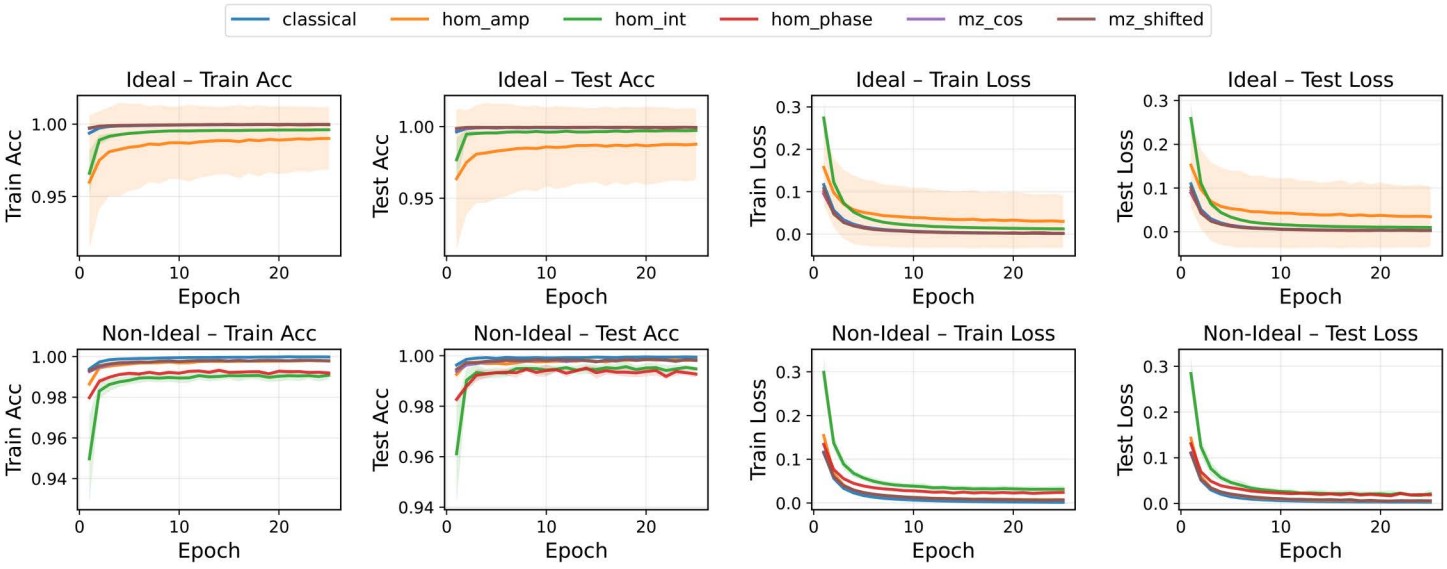

**Fig 3. Training and test accuracy and loss for all pre-activation functions in the MNIST binary classification task (class 0 vs 1).** Ideal (top) and non-ideal (bottom) scenarios are shown. Curves represent the mean over five runs; shaded bands denote the standard deviation.

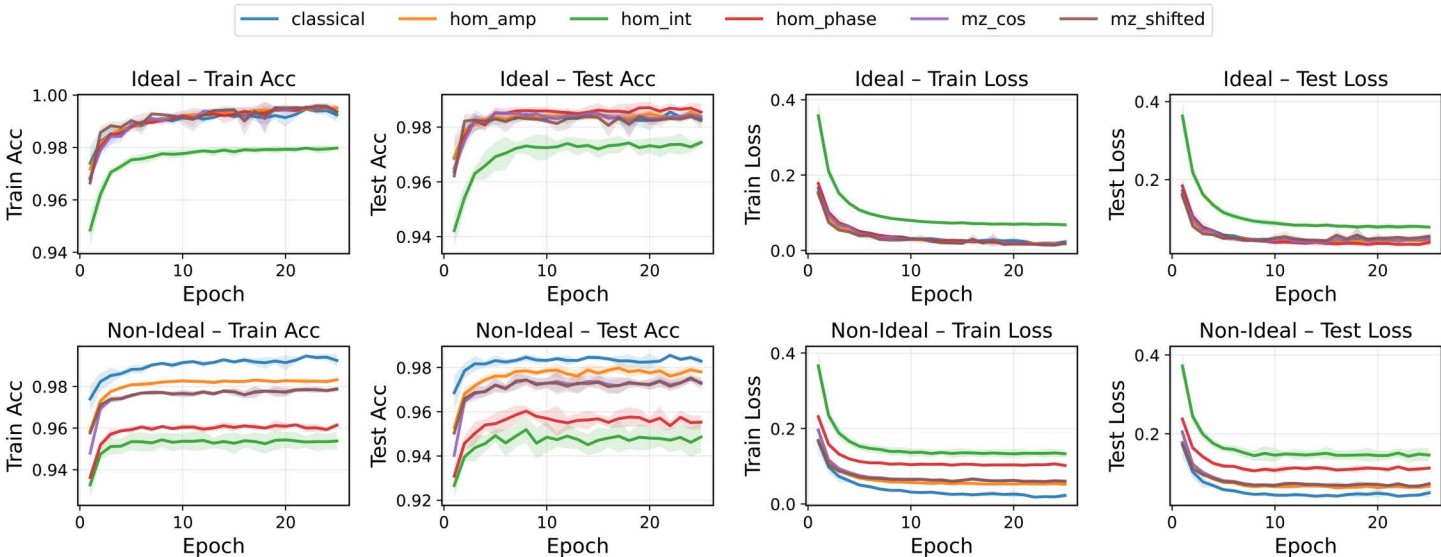

**Fig 4. Training and test accuracy and loss for all pre-activation functions in the FashionMNIST binary classification task (class 0 vs 1).** Ideal (top) and non-ideal (bottom) scenarios are shown. Mean over five runs with standard deviation.

## 5.2 Multiclass classification with a layered neural network

To assess the applicability of quantum optical neurons in more complex scenarios, we extended our experiments to multiclass classification using a layered neural network.

Unlike the single-neuron setup, this architecture consists of three fully-connected QON layers: two hidden layers followed by an output layer. Each neuron in the network uses the same type of pre-activation function $\mathcal{T}$, drawn from the previously defined variants.

The input is a flattened $28 \times 28$ grayscale image from either MNIST or FashionMNIST, normalized to [0, 1] and passed through the QON stack. The first two layers contain 128 and 64 neurons, respectively, each followed by batch normalization and ReLU activation. The final layer outputs 10 unnormalized logits, one per class, which are converted into probabilities via the softmax function. The network is trained using the cross-entropy loss, appropriate for multiclass tasks.

All previously described QON variants were applied within this more advanced architecture. Based on earlier findings, the bias term was again excluded from non-classical variants, as its presence degraded performance. Batch normalization was retained in all layers for consistency and improved stability.

Training was conducted for 25 epochs using the Adam optimizer with a learning rate of 0.01 and a batch size of 64. No further preprocessing was applied beyond pixel normalization. As in the binary setting, each multiclass experiment was repeated five times independently, and all reported curves show the mean value with a shaded band indicating the standard deviation.

For MNIST, the classical network and the HOM amplitude-modulated variant achieved the strongest performance under ideal conditions ([Fig 5](), top row). MZ-based neurons achieved competitive accuracy with stable convergence, though they did not surpass the classical baseline. HOM phase and intensity variants again performed less consistently, with higher variance and less stable loss trajectories.

Under non-ideal conditions ([Fig 5](), bottom row), the HOM amplitude-modulated variant remained the most robust QON variant, while MZ neurons degraded moderately but retained stability. HOM phase and intensity showed the most significant drop in performance.

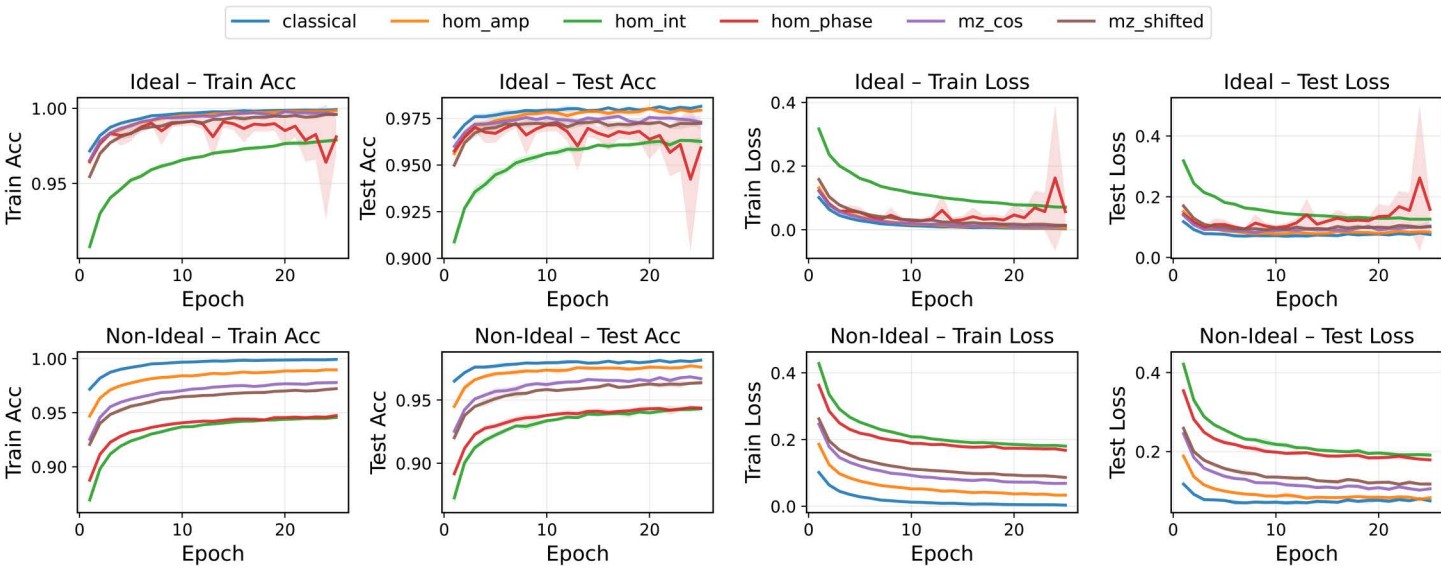

**Fig 5. Multiclass MNIST (10 classes).** Training and test accuracy and loss under ideal (top) and non-ideal (bottom) conditions. Curves represent the mean over five runs with standard deviation.

FashionMNIST further accentuated these differences (Fig 6). The classical model achieved the best performance overall, followed by the amplitude-modulated HOM neuron. MZ neurons maintained reasonable accuracy but showed greater sensitivity to noise in the deeper architecture. HOM phase and intensity models again exhibited the weakest stability.

# 6 Conclusions

Building on the foundational work of Roncallo et al. [25], we extended the study of quantum optical neurons by investigating alternative implementations based on different photon modulation strategies and interferometric architectures. In particular, we introduced a Mach–Zehnder-based design that enables access to both the real and imaginary parts of the scalar product between modulated wave functions, thereby complementing the original formulation based on the simpler HOM interferometer and allowing for additional QON versions. From these physically motivated models, we derived a family of pre-activation functions and systematically evaluated their learning capabilities on a range of classification tasks.

Our experiments show that HOM amplitude-modulated and MZ phase-shifted neurons often achieve strong performance across both binary and multiclass image classification tasks, demonstrating competitive accuracy and stable convergence. By evaluating all models across multiple independent runs and under both ideal and non-ideal conditions, we obtained a comprehensive picture of their robustness and sensitivity to realistic imperfections. This unified analysis indicates that MZ-based neurons retain comparatively stable behavior even in the presence of noise, while HOM amplitude modulation remains competitive—particularly in deeper architectures. In contrast, phase- and intensity-modulated HOM-based variants exhibit reduced stability and a higher susceptibility to noise. The comparison with a classical neuron baseline further shows that several QON variants can approach classical performance while offering physically grounded operations that may translate into practical advantages in photonic implementations.

This study also presents several strengths and limitations that contextualize its findings. Among the strengths, our evaluation is comprehensive: all QON variants are assessed within a unified mathematical and experimental framework, across multiple runs, under both ideal and non-ideal conditions, and in direct comparison with a classical baseline. This provides statistically grounded and physically meaningful insights into the behavior of quantum-inspired optical neurons.

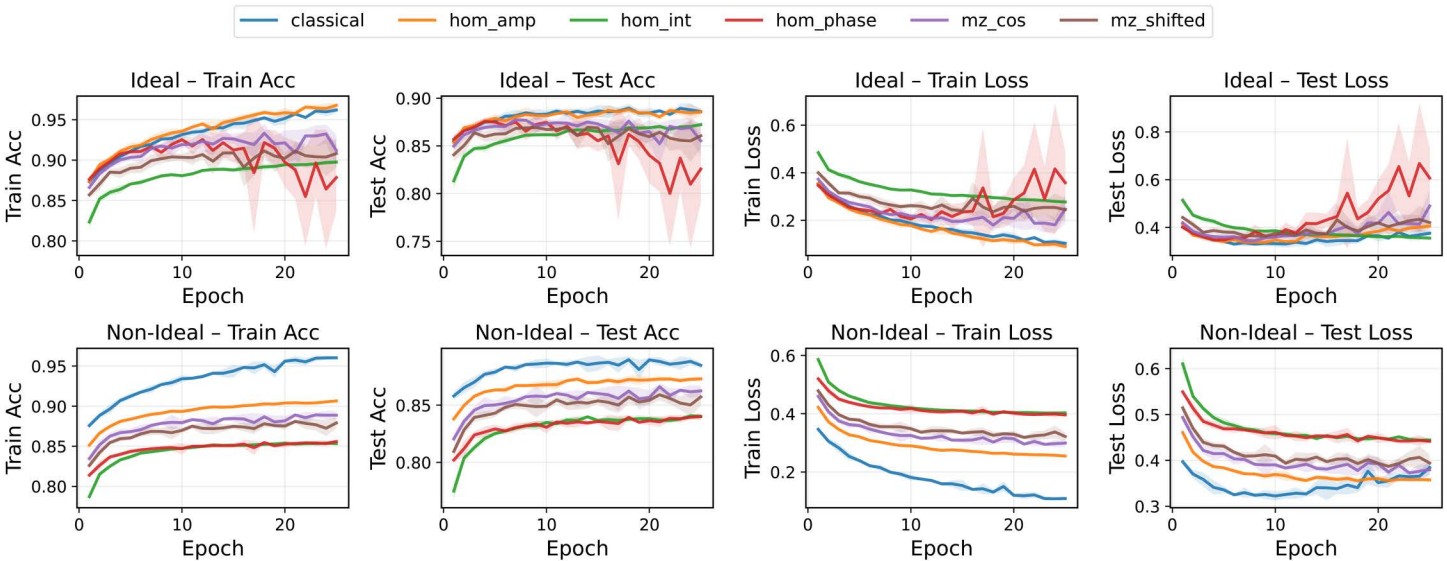

**Fig 6. Multiclass FashionMNIST (10 classes).** Training and test accuracy and loss under ideal (top) and non-ideal (bottom) conditions. Mean over five runs with standard deviation.

However, the analysis is limited by its reliance on software simulations rather than physical implementations, and by the use of relatively shallow architectures and standard image benchmarks. While these controlled settings allow for clear, interpretable comparisons, they do not capture all hardware constraints or the broader range of workloads relevant to photonic accelerators. These limitations point toward natural extensions of this work, including the study of deeper QONNs, hardware-in-the-loop experimentation, and application-specific tasks where optical systems may offer distinctive advantages.

Looking ahead, future work will investigate deeper QON architectures, hybrid optical–electronic designs, and potential hardware realizations. These research directions could pave the way toward integrating QONs as fundamental building blocks for the next generation of energy-efficient and physically grounded AI systems.

## Acknowledgments

The authors would like to thank Giovanni Tempesta and Michele Ciro Di Carlo for their valuable support during the preliminary stages of this work. The authors also acknowledge the use of Grammarly for assisting in the revision of the manuscript's language and style.

## Author contributions

**Conceptualization:** Andrea Andrisani, Gennaro Vessio, Fabrizio Sgobba, Francesco Di Lena, Luigi Amato Santamaria, Giovanna Castellano.

**Formal analysis:** Andrea Andrisani.

**Investigation:** Andrea Andrisani, Gennaro Vessio.

**Methodology:** Andrea Andrisani, Fabrizio Sgobba.

**Software:** Gennaro Vessio.

**Supervision:** Luigi Amato Santamaria, Giovanna Castellano.

**Visualization:** Andrea Andrisani.

**Writing – original draft:** Andrea Andrisani, Gennaro Vessio.

**Writing – review & editing:** Fabrizio Sgobba, Francesco Di Lena, Luigi Amato Santamaria, Giovanna Castellano.

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
