## [Decision Letter · Decision Letter 0]

27 Oct 2025

Dear Dr. Vessio,

Thank you for submitting your manuscript to PLOS ONE. After careful consideration, we feel that it has merit but does not fully meet PLOS ONE’s publication criteria as it currently stands. Therefore, we invite you to submit a revised version of the manuscript that addresses the points raised during the review process.

We look forward to receiving your revised manuscript.

Kind regards,

Uma Maheswari Rajagopalan, Ph.D

Academic Editor

PLOS ONE

Journal Requirements:

3. We note that your Data Availability Statement is currently as follows: [All datasets used in this study are publicly available.]

Please confirm at this time whether or not your submission contains all raw data required to replicate the results of your study. Authors must share the “minimal data set” for their submission. PLOS defines the minimal data set to consist of the data required to replicate all study findings reported in the article, as well as related metadata and methods (https://journals.plos.org/plosone/s/data-availability#loc-minimal-data-set-definition

Reviewers' comments:

Reviewer's Responses to Questions

**Comments to the Author**

1. Is the manuscript technically sound, and do the data support the conclusions?

Reviewer #1: Yes

Reviewer #2: Yes

2. Has the statistical analysis been performed appropriately and rigorously?

Reviewer #1: Yes

Reviewer #2: Yes

3. Have the authors made all data underlying the findings in their manuscript fully available?

Reviewer #1: Yes

Reviewer #2: Yes

4. Is the manuscript presented in an intelligible fashion and written in standard English?

Reviewer #1: Yes

Reviewer #2: Yes

Reviewer #1: This manuscript introduces a family of QON architectures based on Hong–Ou–Mandel (HOM) and Mach–Zehnder (MZ) interferometers, incorporating different photon modulation strategies—phase, amplitude, and intensity. The manuscript is well-written according to the journal scope. I recommend it for publication. However, I will suggest that the author make the following suggestions and major corrections.

1: Rewrite the introduction to grasp the core contributions of the paper. Consider simplifying the language and clearly stating the objectives, methods, and key findings in a more structured manner.

2: Demonstrate the role of the Efficient Neural Computation in detail.

3: The potential impact of the work is hinted at but not clearly articulated. Explicitly state how this research advances the field or solves existing problems.

4: Demonstrate Eqs 7-15 with physical meaning

5: Redraw Figs 4-6 to make them clearer and rewrite their captions.

6: Improve the section “Benchmarking QON architectures: Performance of pre-activation variants”.

7. The author should explain the application of the present problem as the authors mentioned in the literature, and discuss with https: https://doi.org/10.1088/1361-6471/aabb78, https: https://doi.org/10.1088/1361-6471/ab449a.

8: Provide a concise summary of the methods and results.

9: Highlight specific applications or case studies to demonstrate practical relevance.

10: Ensure proper use of references.

11: Highlight the limitations and strengths of the study.

Reviewer #2: Summary

This manuscript presents a comprehensive study of quantum optical neurons (QONs), proposing novel architectures based on Hong–Ou–Mandel (HOM) and Mach–Zehnder (MZ) interferometers. The authors explore various photon modulation strategies—amplitude, intensity, and phase—and implement these as differentiable modules in software. The models are benchmarked on standard image classification tasks (MNIST and FashionMNIST), both in single-neuron and multilayer network configurations. The results demonstrate that certain QON configurations can match or even outperform classical neurons in terms of convergence and stability, highlighting their potential for energy-efficient, scalable AI systems.

Major Concerns

1. Hardware Realizability:

While the models are simulated in software, the manuscript would benefit from a more detailed discussion on the feasibility of implementing these QONs in real photonic hardware. What are the current technological limitations, and how close are we to practical deployment?

2. Scalability and Noise:

The paper claims super-exponential speed-up in inference, but it does not sufficiently address how noise, decoherence, or photon loss might affect performance in physical systems. A discussion on robustness to such imperfections would strengthen the practical relevance.

3. Bias Term in QONs:

The authors note that including a bias term in QONs degrades performance but do not provide a theoretical explanation. This phenomenon deserves further analysis or at least a hypothesis.

Minor Comments and Suggestions

• Notation Consistency:

Some equations (e.g., Eq. 54–55) could benefit from clearer notation, especially when switching between HOM and MZ contexts.

• Figures:

Figures 3–6 are informative, but adding error bars or standard deviation across multiple runs would help assess the statistical robustness of the results.

• References:

The reference list is comprehensive and up-to-date. However, the authors might consider citing recent experimental advances in integrated photonic QML platforms to contextualize their work further.

• Code Availability:

Since the models are implemented in PyTorch, the authors should consider releasing the code to facilitate reproducibility.

Recommendation

Minor Revision

The manuscript presents a significant and timely contribution to the field of quantum-inspired machine learning. With minor revisions addressing the concerns above—particularly regarding hardware feasibility and robustness—the paper will be suitable for publication in PLOS ONE.

**Do you want your identity to be public for this peer review?** For information about this choice, including consent withdrawal, please see our Privacy Policy

Reviewer #1: No

Reviewer #2: No

---

## [Author Response · Author response to Decision Letter 1]

19 Dec 2025

We sincerely thank the Academic Editor and the Reviewers for their time and constructive suggestions, which have helped us to substantially improve the quality and clarity of our manuscript. We carefully addressed all the points raised, revising the text, figures, and references accordingly. A detailed, point-by-point response is provided below, indicating how and where each comment has been addressed in the revised manuscript.

Response to Reviewer #1

This manuscript introduces a family of QON architectures based on HOM and MZ interferometers, incorporating different photon modulation strategies—phase, amplitude, and intensity. The manuscript is well-written according to the journal scope. I recommend it for publication. However, I will suggest that the author make the following suggestions and major corrections.

Response.

We sincerely thank the Reviewer for the positive evaluation of our work and for recognizing its relevance to the journal’s scope. We also appreciate the constructive suggestions provided, which have helped us to further clarify and strengthen the manuscript. Below, we address each comment in detail.

Rewrite the introduction to grasp the core contributions of the paper. Consider simplifying the language and clearly stating the objectives, methods, and key findings in a more structured manner.

Response.

We thank the Reviewer for the suggestion. The Introduction has been restructured to more clearly present the objectives, methodology, and key findings of the work. We now explicitly describe the experimental protocol, including the evaluation under ideal and non-ideal conditions and across multiple runs.

Demonstrate the role of the Efficient Neural Computation in detail.

Response.

We thank the Reviewer for this comment. The role of efficient neural computation in QONs is discussed in detail in Section 3.2, where we analyze the computational cost of each proposed QON architecture by explicitly distinguishing between the inference and training phases.

The potential impact of the work is hinted at but not clearly articulated. Explicitly state how this research advances the field or solves existing problems.

Response.

We have expanded the Introduction to explicitly articulate the potential impact of our findings on future photonic and hybrid photonic–electronic AI systems.

Demonstrate Eqs 7–15 with physical meaning.

Response.

Equations 7–15 provide the explicit mathematical expression of the scalar product between two photon quantum states, as well as its derivatives with respect to the encoded parameters. These expressions are obtained by expanding the quantum states according to the decomposition introduced in Eq. (1) over suitable orthonormal bases, which physically represent the encoding of input data and weights onto the photon wavefronts. Such encoding can be practically realized using spatial light modulators, whose associated orthonormal basis is defined in Eq. (9). To address the Reviewer’s request, we have revised the surrounding text to more clearly emphasize the physical interpretation of these equations and their connection to the optical encoding process.

Redraw Figs 4–6 to make them clearer and rewrite their captions.

Response.

In the revised version, Figures 3–6 have been completely redrawn to improve visual clarity and readability. The updated figures now include averaged performance across five independent runs with shaded bands representing the standard deviation, providing a clearer and statistically robust comparison among the different QON variants. Captions have also been rewritten to more precisely describe the experimental setting, the meaning of the error regions, and the performance trends.

Improve the section “Benchmarking QON architectures: Performance of pre-activation variants”.

Response.

We thank the Reviewer for this useful remark. The section has been substantially improved. In particular, we (i) incorporated the analysis over multiple independent runs, (ii) added ideal and non-ideal experimental conditions, (iii) updated all figures to include mean performance with standard-deviation bands, and (iv) expanded the discussion to highlight differences in stability, robustness, and convergence across QON variants and relative to the classical baseline.

The author should explain the application of the present problem as the authors mentioned in the literature, and discuss with https://doi.org/10.1088/1361-6471/aabb78, https://doi.org/10.1088/1361-6471/ab449a.

Response.

We thank the Reviewer for this suggestion. In the revised Introduction, we have expanded the discussion on physically inspired quantum computing paradigms and cited recent works based on Bose–Einstein condensate technologies, including https://doi.org/10.1088/1361-6471/ab449a

, which we consider the most recent and representative among the two suggested references.

Provide a concise summary of the methods and results.

Response.

We thank the Reviewer for this comment. The abstract has been revised to provide a more concise and informative summary of both methods and results.

Highlight specific applications or case studies to demonstrate practical relevance.

Response.

We have added concrete examples of application domains—such as autonomous vision, wearable sensing, remote sensing, and scientific imaging—where the properties of QONs are particularly advantageous.

Ensure proper use of references.

Response.

We have carefully rechecked all references in the revised manuscript. We remain fully available to collaborate with the editorial and production teams to make any additional formatting adjustments that may be required during the typesetting process.

Highlight the limitations and strengths of the study.

Response.

We thank the Reviewer for this suggestion. The Conclusion section has been expanded to discuss the strengths and limitations of the study explicitly.

Response to Reviewer #2

This manuscript presents a comprehensive study of quantum optical neurons, proposing novel architectures based on HOM and MZ interferometers. The authors explore various photon modulation strategies—amplitude, intensity, and phase—and implement these as differentiable modules in software. The models are benchmarked on standard image classification tasks (MNIST and FashionMNIST), both in single-neuron and multilayer network configurations. The results demonstrate that certain QON configurations can match or even outperform classical neurons in terms of convergence and stability, highlighting their potential for energy-efficient, scalable AI systems.

Response.

We sincerely thank the Reviewer for the positive and encouraging evaluation of our work. We genuinely appreciate the acknowledgment of our contributions to quantum-inspired neural computation and the recognition of the proposed QON architectures as a relevant and timely development toward energy-efficient, scalable AI models. We are also grateful for the constructive comments, which have helped us improve the clarity and completeness of the manuscript.

Hardware Realizability:

While the models are simulated in software, the manuscript would benefit from a more detailed discussion on the feasibility of implementing these QONs in real photonic hardware. What are the current technological limitations, and how close are we to practical deployment?

Response.

We thank the Reviewer for this important comment. We have added a new dedicated section (Section 4) that discusses the feasibility of implementing QONs in real photonic hardware.

Scalability and Noise:

The paper claims super-exponential speed-up in inference, but it does not sufficiently address how noise, decoherence, or photon loss might affect performance in physical systems. A discussion on robustness to such imperfections would strengthen the practical relevance.

Response.

To address concerns about robustness and scalability under realistic conditions, we have added a new subsection (Section 3.1.6) explicitly devoted to noise and decoherence. In addition, new simulations under non-ideal conditions have been included in Section 5, enabling us to quantitatively assess the robustness of the proposed QON variants with respect to accuracy, convergence, and stability. Considerations on photon loss and its effect on effective computational times and measurement statistics have also been added in Section 4, in the context of hardware feasibility.

Bias Term in QONs:

The authors note that including a bias term in QONs degrades performance but do not provide a theoretical explanation. This phenomenon deserves further analysis or at least a hypothesis.

Response.

We thank the Reviewer for this valuable comment. In the revised manuscript (Section 5.1), we have added a hypothesis to explain why the inclusion of a bias term tends to degrade the performance of quantum optical neurons. We hypothesize that this behavior is primarily related to optimization difficulties rather than to a fundamental limitation of QONs. In particular, the proposed quantum-inspired pre-activation functions are highly nonlinear and often involve normalized or bounded quantities, whereas the bias term is purely linear and additive. This mismatch can alter the geometry of the loss landscape, making the overall optimization problem more difficult and leading to less stable convergence. We emphasize that this explanation is currently a hypothesis supported by experimental observations, and that a deeper theoretical analysis represents an interesting direction for future work.

Notation Consistency:

Some equations (e.g., Eq. 54–55) could benefit from clearer notation, especially when switching between HOM and MZ contexts.

Response.

Done. We have revised the notation in the indicated equations by explicitly distinguishing between HOM- and MZ-based formulations, introducing appropriate subscripts where needed.

Figures:

Figures 3–6 are informative, but adding error bars or standard deviation across multiple runs would help assess the statistical robustness of the results.

Response.

We thank the Reviewer for this valuable suggestion. In the revised manuscript, Figures 3–6 have been fully updated. Each experiment is now repeated over five independent runs, and the plots report the mean performance together with a shaded band indicating the standard deviation, thereby improving the statistical robustness of the comparison. In addition, for each benchmark, we now include the corresponding non-ideal scenario that incorporates noise effects. The main text has been updated to explicitly address the variability observed across runs and the impact of non-ideal conditions. All figure captions have been updated accordingly.

References:

The reference list is comprehensive and up-to-date. However, the authors might consider citing recent experimental advances in integrated photonic QML platforms to contextualize their work further.

Response.

To further contextualize our work with respect to recent experimental advances in photonic quantum machine learning, we have added Ref. [35]: “Experimental realization of a quantum image classifier via tensor-network-based machine learning” (Wang et al., 2021).

Code Availability:

Since the models are implemented in PyTorch, the authors should consider releasing the code to facilitate reproducibility.

Response.

We thank the Reviewer for this helpful suggestion. To support transparency and reproducibility, we have made the complete PyTorch implementation publicly available on GitHub. The repository can be accessed at:

https://github.com/gvessio/quantum-optical-neurons

Recommendation: Minor Revision

The manuscript presents a significant and timely contribution to the field of quantum inspired machine learning. With minor revisions addressing the concerns above, particularly regarding hardware feasibility and robustness, the paper will be suitable for publication in PLOS ONE.

Response.

We sincerely thank the Academic Editor for the positive evaluation and for recommending the manuscript for publication pending minor revisions. We have carefully addressed all points raised in the decision letter and reviewers’ reports. We believe these revisions have improved the clarity and completeness of the manuscript, and we hope it now meets the journal’s publication standards.

---

## [Editor Report · Decision Letter 1]

8 Jan 2026

Modeling and benchmarking quantum optical neurons for efficient neural computation

PONE-D-25-47255R1

Dear Dr. Vessio,

We’re pleased to inform you that your manuscript has been judged scientifically suitable for publication and will be formally accepted for publication once it meets all outstanding technical requirements.

Kind regards,

Uma Maheswari Rajagopalan, Ph.D

Academic Editor

PLOS One
---

## [Editor Report · Acceptance letter]

PONE-D-25-47255R1

PLOS One

Dear Dr. Vessio,

I'm pleased to inform you that your manuscript has been deemed suitable for publication in PLOS One. Congratulations! Your manuscript is now being handed over to our production team.

Kind regards,

on behalf of

Dr. Uma Maheswari Rajagopalan

Academic Editor

PLOS One